# A Comparative Study of Selected Gut Bacteria Abundance and Fecal pH in Bodybuilders Eating High-Protein Diet and More Sedentary Controls

**DOI:** 10.3390/nu13114093

**Published:** 2021-11-16

**Authors:** Joanna Szurkowska, Jakub Wiącek, Konstantinos Laparidis, Joanna Karolkiewicz

**Affiliations:** 1Department of Food and Nutrition, Poznan University of Physical Education, 61-871 Poznań, Poland; szurkowskajoanna@gmail.com (J.S.); wiacek@awf.poznan.pl (J.W.); 2Department of Physical Education and Sports Science, School of Physical Education & Sport Sciences, Democritus University of Thrace University Campus, 69100 Komotini, Greece; lapco@phyed.duth.gr

**Keywords:** gut microbiota, bodybuilders, high-protein diet

## Abstract

Bodybuilders tend to overeat their daily protein needs. The purpose of a high-protein diet is to support post-workout recovery and skeletal muscle growth; however, its exact impact on gut microbiota still remains under investigation. The aim of this study was to assess the differences in selected gut bacteria (*Faecalibacterium prausnitzii*, *Akkermansia muciniphila*, *Bifidobacterium* spp., and *Bacteroides* spp.) abundance and fecal pH between the group of amateur bodybuilders and more sedentary control group. In total, 26 young healthy men took part in the study, and their daily nutrients intake was measured using a dietary interview. Real-time PCR was used to assess the stool bacteria abundance. Both groups reported fiber intake within the recommended range, but bodybuilders consumed significantly more protein (33.6% ± 6.5% vs. 22% ± 6.3%) and less fat (27.6% ± 18.9% vs. 36.4% ± 10%) than controls. Study results showed no significant differences in terms of selected intestinal bacteria colony forming unit counts. Significantly higher fecal pH in the bodybuilders’ fecal samples was observed in comparison to the control group 6.9 ± 0.7 vs. 6.2 ± 0.7. Gut microbiota composition similarities could be a result of appropriate fiber intake in both groups.

## 1. Introduction

Bodybuilding aims to develop muscle mass, maintain symmetry, and keep the body fat levels as low as possible. To achieve these goals, bodybuilders use specific diets and resistance training plans. Building muscle mass requires caloric intake above the level of energy expenditure and it usually takes place in the off-season (after the contest). Another important element affecting skeletal muscle hypertrophy during this phase is the increased protein supply throughout the day. Bodybuilders should consume between 1.6 and 2.2 g of protein per every kilogram of the body mass (g/kg b.w.) [1]. Together with caloric surplus, it creates an anabolic environment for post-workout recovery and muscle protein synthesis (MPS) [2]. On the other hand, the pre-contest diet of bodybuilders is low in calories. Caloric deficit is necessary to activate catabolic pathways involved in loosing body fat mass. Contest preparation diets should provide recommended amounts of protein to avoid decrease in muscle mass. However, it has been reported that bodybuilders often exceed the recommended amounts of protein, as it reaches levels of 4.3 g/kg b.w./day (men) and 2.8 g/kg b.w./day (women) [3]. High-protein diets and protein supplements allow bodybuilders to increase muscle mass, but its effects on the gut microbiota needs further research. 

The variety and number of health-promoting intestinal bacteria depend on such factors as the level of physical activity and diet quality. Altering protein or carbohydrates and dietary fiber intake in bodybuilders may change the gut bacteria abundance and composition and influence host metabolism and immune function [4]. Some of the gut bacteria are able to synthesize amino acids de novo and it affects the level of nitrogen in the body alongside with dietary protein intake. Inadequate fiber intake may enhance proteolytic fermentation due to decreased availability of fermentable carbohydrates. As a consequence, it may result in potentially harmful metabolites production increase [5]. Moreover, microbial fermentation of undigested peptides is another important source of molecules that contribute to the body’s amino acid pool. It also affects inflammation in the host’s tissues, which results in tissue permeability modulation [6]. 

Optimal composition of the gut microbiota is important for nitrogen balance and muscle protein synthesis, but also for muscle glycogen storage and oxidative stress management [7]. Intestinal bacteria promote carbohydrate fermentation and short-chain fatty acids (SCFAs) production. SCFAs (e.g., butyric acid, propionic acid, acetic acid) enhance the intestinal epithelial membrane, support absorption of electrolytes, and regulate glucose metabolism in skeletal muscles [8]. Gut microbiota imbalances (dysbiosis) may cause a number of gastrointestinal disorders, having a negative impact on sports results and overall health and wellbeing. It is a bidirectional connection in which exercise induces a health-promoting shift in the microbial environment that affect the host’s energy metabolism, oxidative stress balance, and immune system functioning. However, excessive training frequency and intensity may lead to negative changes within intestinal microbiota and limit muscle recovery and adaptations. 

There are clear indications that sports training changes the gut microbiota. Nevertheless, in a human study, it is difficult to distinguish the effect of diet from that of exercise. We attempted to verify whether a high-protein diet used by bodybuilders modulates the abundance of health-promoting gut bacteria. The main objective of this study was to compare the targeted gut microbiota and stool pH between amateur bodybuilders on a high-protein diet and individuals who did not practice resistance training and consumed a balanced diet. 

## 2. Materials and Methods

### 2.1. Participants 

The study group consisted of 26 young healthy men aged 22–28 years. There were two subgroups: amateur bodybuilders (*n* = 11) and controls of a similar age (*n* = 15). Bodybuilders’ mean experience was 5 ± 3 years of training with a frequency of 5 trainings per week (total weekly training time: minimum of 7.5 h). Members of the sport club TKKF Winogrady took part in this study during the muscle-building phase of a contest preparation diet. The control group consisted of students of the Poznan University of Physical Education, who reported low or medium levels of physical activity and a balanced diet. 

Before entering the study, participants were interviewed on their training, nutrition, and dietary supplementation history and habits. The study inclusion criteria were as follows: voluntary written consent, age of 18 or older, good overall health, no gastrointestinal or respiratory infections in the last 4 weeks, and no injuries followed by inflammation in the last 4 weeks. We excluded from the study participants who used antibiotics, proton pump inhibitors (PPIs), probiotics, prebiotics, androgenic-anabolic steroids, oral antimicrobial agents, or travelled to countries with a different climate and habitual diet during the last 4 weeks before the study. We also excluded subjects who underwent hospitalization in the previous month. Information regarding health status, medical history, and total exercise workload was assessed using a questionnaire on the day of the survey.

The study design met the criteria of Ethics Guidelines of the Declaration of Helsinki. It was approved by the Local Ethics Committee at the Poznan University of Medical Sciences, reference numbers no.173/16. Only subjects who signed the consent could take part in the study. Data collection was conducted according to the Helsinki declaration for biomedical research on human subjects. 

Daily intake of energy and nutrients was assessed using dietary interviews, based on participants’ three day nutrition (2 working days and 1 weekend day). Study subjects self-reported the estimated meal composition and weight of the used products in the diary. The amount of nutrients in participants’ diets was evaluated using the NUVERO application (Poland). An information on dietary supplements consumption (e.g., whey protein concentrate) was also recorded. 

Dual-Energy X-ray Absorptiometry (DXA; GE Healthcare Lunar Prodigy Advance; GE Medical Systems, Milan, Italy) was used to perform an analysis of participants’ body composition. The subjects from both groups rested for over 24 h before taking part in the study. Before the examination started, subjects received clear instructions on the procedure rules and order. 

### 2.2. Stool Sample Collection

Selected gut bacteria abundance analysis and fecal pH measurement required stool sample collection. The subjects were asked to bring it to the laboratory as quickly as possible (in under 24 h). The KyberKompaktPRO (Institute of Microecology) protocol was used to properly take the samples (3/4 of volume of 150-mL container, pieces of stool from up to 8 locations). 

A QIAamp Fast DNA Stool Mini Kit (QIAGEN) was used to prepare selected gut bacteria DNA from feces. This procedure was conducted following the manufacturer’s instructions. The samples were frozen and left in the freezer until further analysis. Real-Time PCR (ABI 7300; ThermoFisher Scientific, Waltham, MA, USA) was used to perform anaerobic gut bacteria abundance analysis. Table 1 shows the selected primers (ThermoFisher Scientific) needed to assess the counts of *Faecalibacterium prausnitzi*, *Akkermansia muciniphila* of the genus *Akkermansia*, *Bifidobacterium* spp. of the genus *Actinobacteria*, and *Bacteroides* spp. of the genus Bacteroidetes.

The real-time PCR results were recalculated to bacteria count per gram. The amplification efficacy in all assays was higher than 90%. The standard curve showed a linear range across at least 5 logs of DNA concentrations with a correlation coefficient >0.99. The lowest detection limits of all assays were as low as 10–100 copies of specific bacterial 16S rDNA per reaction, which corresponds to 104–105 copies per gram of wet-weight feces. Knowing the values of the standards and their C(t) (cycle threshold), the obtained data were converted using the right coefficients. The standards used in the study are listed in Table 2.

The lowest value for bacteria detectability was 102 colony forming units (CFU) per gram of feces. To simplify the calculation, any results under this level were set as “0” in the statistical analysis. After the conversion of microbiota analysis results, it was shown as the decimal logarithm (Log10). The procedures of the gut bacteria real-time PCR analysis were performed according to the instructions given by Institute of Microecology in Herborn, Germany. Table 3 shows the reference values for the selected bacteria.

### 2.3. Statistical Analysis 

The obtained test results were examined using statistic tools to show the differences between observed groups (STATISTICA 13.0; StatSoft Inc., Palo Alto, CA, USA). The results presentation includes mean values with standard deviations (± SD) and/or medians (Me) with Q1 and Q3 quartiles. Since the obtained data violated normality and demonstrated heterogeneous variability, non-parametric tests were used. The significance of differences between the bodybuilder and control group outcomes was assessed using the Mann–Whitney test.

## 3. Results

### 3.1. Body Composition Analysis

The performed analysis demonstrated a significant difference in body mass (*p* < 0.05) and fat-free mass (*p* < 0.05) between the groups (Table 4).

### 3.2. Nutrients Intake Analysis

The dietary survey analysis presented significant differences in protein (*p* < 0.05) and fat (*p* < 0.01) consumption between the examined groups (Table 5). Figure 1 shows the proportions of nutrients consumed by the study participants. Bodybuilders ate less calories from fat and more calories from protein than controls. However, there was no significant difference in protein consumption expressed as g/kg b.w. The mean fiber intake in both groups reached the levels recommended for daily consumption. 

### 3.3. Stool Samples Analysis

The performed analysis showed no statistically significant differences in the examined gut bacteria abundance expressed as decimal logarithm of CFU per gram of feces. This study revealed a significant difference in fecal pH values (*p* < 0.05) between bodybuilders and the control group (Table 6).

## 4. Discussion

This study’s results showed that bodybuilders’ high-protein diet with fiber intake on the recommended level, in comparison to a standard diet, did not promote any significant changes in the abundance of the examined health-promoting gut bacteria. However, the bodybuilders’ fecal pH was higher than that of the control group. The results of studies carried out by other researchers showed that long-term dietary changes have a big impact on the dominant gut bacteria species, while changes of shorter duration play only a temporary role [9,10]. However, there is still insufficient information on this issue, and it is limited to people with protein intake that exceed the nutritional requirements by 2–4 fold. 

Our study was performed during the mass-gaining (muscle-building) phase of the bodybuilders’ diet. During this period, bodybuilders focus on muscular hypertrophy and it often leads to an increase in adipose tissue deposition. Therefore, there were notable differences in total body mass (mean ± SD: 96.4 ± 8.9 vs. 83.4 ± 13.2; *p* < 0.005) and fat-free body mass (mean ± SD: 80.6 ± 8.9 vs. 69.7 ± 6.4; *p* < 0.005) between bodybuilders and the control group (Table 4). The systematic review of 18 manuscripts on bodybuilding showed that the mean percentage of adipose tissue in this group of athletes reaches 12.1 ± 2.5% in the off-season, while contest preparation requires a body-fat level of about 6% (for men) [3]. In our study, bodybuilders’ mean percentage of body fat mass was 14% and did not differ significantly from the control group men (Table 4). 

The analysis performed in our study showed that male bodybuilders consumed on average 3516 kcal per day, with 38.8% of that energy coming from carbohydrates, 33.6% from protein, and 27.6% from fats. We found no significant difference in dietary intake between the bodybuilders and control group men in calorie intake per day, percentage of carbohydrates in energy intake, and fiber intake (Table 5). Bodybuilders’ protein intake (%) was significantly higher than that of the control group (Mean ± SD: 33.6% ± 6.5% vs. 22% ± 6.3%; *p* < 0.05) and exceeded the recommended 25–30% of daily energy intake from protein [11]. Still, the mean protein consumption, expressed as g/kg b.w., in the bodybuilder group fitted the recommended ranges and was not as high as expected and did not differ significantly from that of the control group (mean ± SD: 2.1 ± 1.5 vs. 1.7 ± 1.0%; *p* < 0.39). A protein intake level that is adjusted to the body’s needs is essential for maximizing muscle adaptations, increasing muscular hypertrophy, and gaining strength [12]. Bodybuilders consuming a high-protein diet focus on the sources of branched-chain amino acids, especially leucine (e.g., whey, lean meat, eggs). Together with its metabolite, β-hydroxy-β-methylbutyrate (HMB), it may affect the gut environment and nutrient metabolism [13,14]. On the other hand, excessive protein intake increases the abundance of protein-fermenting bacteria, such as *Clostridium, Bacillus, Staphylococcus*, and other species of the *Proteobacteria* family [15]. There are indications from animal studies that increasing the ratio of energy intake from protein to that from carbohydrates and fats may affect microbial composition and protein fermentation as well as its toxic product levels [16]. Moreover, a protein-enriched diet may cause a reduction of carbohydrate-fermenting bacteria, such as *Bacteroides, Lactobacillus, Bifidobacterium, Prevotella, Ruminococcus, Roseburia*, and *Faecalibacterium* [17]. The metabolic processes following the fermentation of incompletely digested protein in the colon may lead to toxic metabolite production (e.g., ammonia, biogenic amines, indole compounds, and phenols). These factors are known enhancers of inflammation and tissue permeability [18]. It is a potential cause of intestine tissue damage and metabolic, immune, and neurological disruptions.

Although increased protein intake was observed in bodybuilders in the study, there was no significant difference in the abundance of selected beneficial bacteria (Table 6). Although the control group consumed more fiber, both tested groups met the criteria for recommended fiber intake. High protein intake’s effects on the microbiome could have been attenuated by appropriate carbohydrate and fiber intake [7]. It stands in accordance with the conclusions of other authors saying that there is no significant difference in gut microbiota diversity between bodybuilders who consume a high-protein/low-fiber diet and the general population [19]. However, the authors of this paper emphasize that the relationship between diet, exercise, and gut microbiota is still very poorly explained.

The pH along the human colon usually varies from 5 to 7, and it depends on fermentation processes, secretion of bicarbonate by colonic epithelial cells, and absorption of microbial metabolites by host epithelial cells [20]. In this conducted study, significantly higher fecal pH levels (mean ± SD: 6.9 ± 0.7 vs. 6.2 ± 0.7; *p* < 0.05) were observed in bodybuilders compared to the male control group (Table 6). High levels of animal-based dietary protein intake may play a role in increasing fecal pH values in bodybuilders, as proteolytic putrefactive bacteria are able to produce alkaline metabolites [21,22]. There are indications that altered pH in the gut may change the microbiota composition and its metabolism (e.g., production of short-chain fatty acids), but in our study, there were no statistically significant differences between the groups in the tested fecal bacteria abundance. In both groups, a reduced abundance of bacteria, such as *Bifidobacterium* spp., *Faecalibacterium prausnitzii* spp., and *Akkermansia muciniphila* spp., was noted (Table 6). Our observations seem to confirm the findings of other authors showing an increasing deficit of F. prausnitzii and A. municiniphia in the human intestine over the past few years [23]. It also confirms other authors’ findings saying that the gut abundance of *Bacteroidetes* spp. is generally not affected by higher protein intake [16,24].

## Figures and Tables

**Figure 1 nutrients-13-04093-f001:**
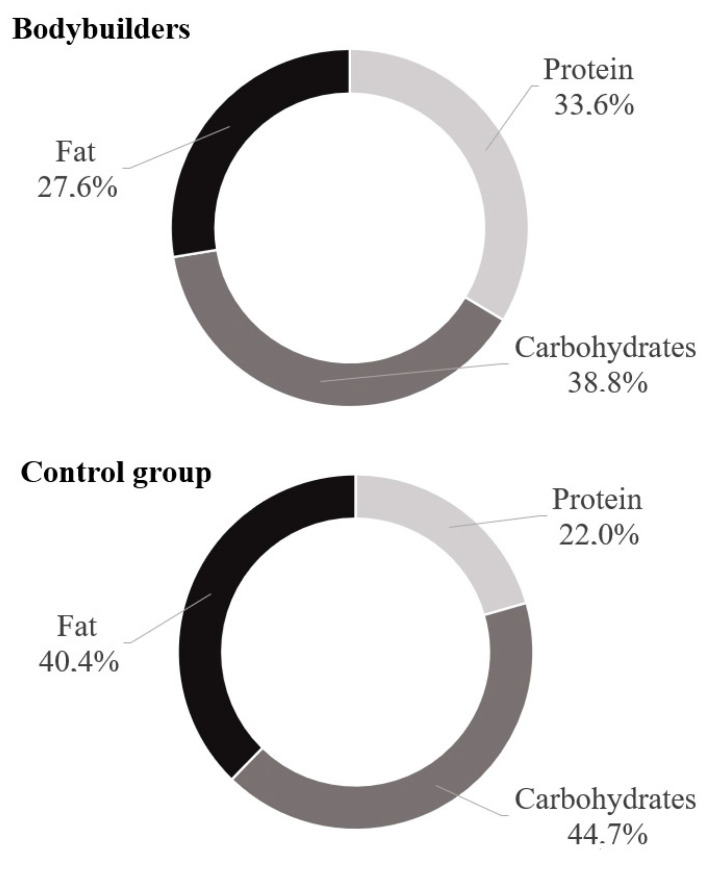
Comparison of daily nutrient intake of bodybuilders and controls.

**Table 1 nutrients-13-04093-t001:** Primers used for the determination of different bacteria.

Name	Product Description	Sequence
Praus-F480	*Faecalibacterium prausnitzii* forward starter	CAGCAGCCGCGGTAAA
Praus-R631	*Faecalibacterium prausnitzii* reverse starter	CTACCTCTGCACTACTCAAGAAA
Akk.muc-F	*Akkermansia muciniphila* starter forward	CAGCACGTGAAGGTGGGGAC
Akk.muc-R	*Akkermansia muciniphila* starter reverse	CCTTGCGGTTGGCTTCAGAT
F-Bifid09c	*Bifidobacterium* spp. forward starter	CGGGTGAGTAATGCGTGACC
R-Bifid06	*Bifidobacterium* spp. reverse starter	TGATAGGACGCGACCCCA
Bacter11	*Bacteroides* spp. forward starter	CCTWCGATGGATAGGGGTT
Bacter08	*Bacteroides* spp. starter reverse	CACGCTACTTGGCTGGTTCAG
Uni-F340	Universal forward starter	ACTCCTACGGGAGGCAGCAGT
Uni-R514	Universal reverse starter	ATTACCGCGGCTGCTGGC

**Table 2 nutrients-13-04093-t002:** Standards applied for the determination of different microorganisms.

Name	Among of DNA (Copies/mL)	Product Description
*Bifidobacterium infantis* DNA	5 × 10⁸	Standard in identification of *Bifidobacterium* spp., isolated from *Bifidobacterium infantis*
*Bacteroides fragilis* DNA	2 × 10⁹	Standard in identification of *Bacteroides* spp., isolated from *Bacteroides fragilis*
*Faecalibacterium prausnitzii* DNA	7.8 × 10⁸	Standard in identification of *Faecalibacterium prausnitzii*
*Akkermansia muciniphila* DNA	3.9 × 10⁸	Standard in identification of *Akkermansia muciniphila*

**Table 3 nutrients-13-04093-t003:** Reference values for selected bacteria.

Species [Genus]	Standard (Log10 CFU/g Feces)	Method
*Bifidobacterium* spp.	≥8	Real-time PCR
*Bacteroides* spp.	≥9	Real-time PCR
*Faecalibacterium prausnitzii*	≥9	Real-time PCR
*Akkermansia muciniphila*	≥8	Real-time PCR

**Table 4 nutrients-13-04093-t004:** Comparison of basic characteristics of body composition.

	Bodybuilder Group (*n* = 11)		Control Group (*n* = 15)		Mann- Whitney (*p*-Value)
	Mean ± SD	Median Q1 ÷ Q3	Mean ± SD	Median Q1 ÷ Q3	
Age (years)	27 ± 6	25 23 ÷ 28	29 ± 8	24 22 ÷ 37	NS
Height [cm]	182.0 ± 6.3	181.5 179.3 ÷ 185	181.7 ± 4.4	182 179 ÷ 185	NS
Body mass [kg]	96.4 ± 8.9	96.8 93.8 ÷ 103	83.4 ± 13.2	76.6 72.4 ÷ 99.8	0.0023 *
Body fat mass [%]	14.0 ± 4.5	14.6 9.5÷18.2	15.3 ± 7.7	15.8 6.6 ÷ 20.7	NS
Body fat mass [kg]	13.2 ± 4.2	13.7 7.9 ÷ 16.7	13.5 ± 8.5	11.6 5 ÷ 21.1	NS
Fat-free mass [kg]	80.6 ± 8.9	81.1 74 ÷ 87.2	69.7 ± 6.4	70.6 63.2 ÷ 74.0	0.0035 *

SD—standard deviation; Q1—lower quartile; Q3—upper quartile; * *p* < 0.005—Statistical significance; NS—no significant differences.

**Table 5 nutrients-13-04093-t005:** Comparison of daily intake of energy [kcal], protein [%; g/kg b.w.], carbohydrates [%], fats [%], and fiber [g] of bodybuilders and the control group.

	Bodybuilders (*n* = 11)		Control Group (*n* = 15)		Mann-Whitney (*p*-Value)
	Mean ± SD	Median Q1 ÷ Q3	Mean ± SD	Median Q1 ÷ Q3	
Energy [kcal]	3516 ± 1433	3032 2685 ÷ 3951	2882 ± 1422	2640 2038 ÷ 3233	NS

Protein [%]	33.6 ± 6.5	34.3 29.2 ÷ 39.2	22 ± 6.3	21.4 18.0 ÷ 24.0	0.0493 *

Protein [g/kg b.w.]	2.1 ± 1.5	2.4 0.0 ÷ 3.1	1.7 ± 1.0	1.8 0.7 ÷ 2.4	NS
Carbohydrates [%]	38.8 ± 14.8	43.2 38.3÷45.6	44.7 ± 14.2	41.7 36.9 ÷ 48.0	NS

Fat [%]	27.6 ± 18.9	21.1 16.0 ÷ 27.4	40.4 ± 10.0	36.4 35.3 ÷ 41.6	0.0002 **

Fiber [g]	29.4 ± 11.8	26.7 25.0 ÷ 33.0	33.8 ± 24.9	31.6 15.5 ÷ 41.7	NS


SD—standard deviation; Q1—lower quartile; Q3—upper quartile; * *p* < 0.05; ** *p* < 0.001—Statistical significance; NS—no significant differences; g/kg b.w.—grams per kilogram of body weight.

**Table 6 nutrients-13-04093-t006:** Comparison of the targeted stool bacteria and fecal pH values of bodybuilders and the control group.

	Bodybuilders (*n* = 11)		Control Group (*n* = 15)		Mann-Whitney (*p*-Value)
	Mean ± SD	Median Q1 ÷ Q3	Mean ± SD	Median Q1 ÷ Q3	
*Bifidocacterium* spp.	6.4 ±0.4	6.3 8.9 ÷ 9.3	7.0 ± 0.6	7.0 8.3 ÷ 9.3	NS
*Bacteroides* spp.	9.0 ± 0.4	9.0 8.9 ÷ 9.3	8.8 ± 0.6	8.8 8.3 ÷ 9.3	NS
*F. prausnitzii*	8.3 ± 0.6	8.5 8.0 ÷ 8.7	7.9 ± 0.6	7.9 7.3 ÷ 8.3	NS
*A. Muciniphila*	6.0 ± 1.4	6.0 4.8 ÷ 7.3	5.6 ± 2.1	6.6 4.7 ÷ 7.0	NS
Fecal pH	6.9 ± 0.7	7.0 6.0 ÷ 7.5	6.2 ± 0.7	6.5 5.5 ÷ 6.5	0.0322 *

SD—standard deviation; Q1—lower quartile; Q3—upper quartile; * *p* < 0.05—Statistical significance; NS—no significant differences.

## Data Availability

The datasets generated and/or analyzed during the current study are available from the corresponding author on reasonable request.

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
