# Peer review of "A Comparative Study of Selected Gut Bacteria Abundance and Fecal pH in Bodybuilders Eating High-Protein Diet and More Sedentary Controls"

_nutrients, 2021, doi:10.3390/nu13114093_

Round 1

Reviewer 1 Report

The authors of "A Comparative Study of Selected Gut Bacteria Abundance and Fecal pH in Bodybuilders Eating High-Protein Diet and More Sedentary Controls" discusses possible differences in gut bacteria population and fecal pH among athletes who consume a diet rich in protein and people with a normal diet.

The article is easy to read and the data and language are clear. However, the novelty of the results is not clear, it seems that data similar to those presented here have already been published by other authors, so I encourage the authors of this manuscript to clarify and point out the importance of these results and their contribution. to the current knowledge of the subject.

Also, here are some things that authors should include:

- At Materials and Methods section:

I suppose that all external factors that could modify the microbiota should be considered and based on that, the participants will be included or excluded from the study. This idea must be expressly indicated in point 2.1 Participants. Furthermore, I ask the authors why having travelled to tropical countries is an exclusion criterion.

Point 2.2. Stool sample collection should indicate also “and PCR assays “. Here, authors should explain in more detail how they have done the conversion from “the number of sequence´s copies amplified” to “bacteria count per gram”. Usually, in a real time PCR reaction, data are obtained as C(t) (cycle threshold), that is the number of cycles required for the fluorescent signal to cross the threshold. So, please clarify and explain.

- In the results section: why did the authors choose the species of bacteria indicated in table 3 and not others? Are they all carbohydrate fermenting bacteria? Also Akkermansia? Why weren't protein fermenting bacteria also measured? It would be interesting to measure both groups of bacteria, carbohydrate and protein fermenting bacteria and establish a balance between beneficial and harmful bacteria. I suggest that the authors do so.

- Presentation of results: authors only use tables to present their data. Could they use any other system, such as diagrams? Results are more attractive for the readers with those type of presentation.

Author Response

Answers to Reviewer 1

- The article is easy to read and the data and language are clear. However, the novelty of the results is not clear, it seems that data similar to those presented here have already been published by other authors, so I encourage the authors of this manuscript to clarify and point out the importance of these results and their contribution. to the current knowledge of the subject.

The number of studies on gut microbiota in athletes is still insufficient to formulate specific conclusions. As the bodybuilders often consume too much protein and not enough fiber, the aim of this study was to assess the differences between high protein diet with resistance training and normal diet.  In this study, we have shown that adequate fiber intake may be a protective factor for health-promoting gut bacteria, as the gut microbiota did not differ significantly between groups.

- I suppose that all external factors that could modify the microbiota should be considered and based on that, the participants will be included or excluded from the study. This idea must be expressly indicated in point 2.1 Participants. Furthermore, I ask the authors why having travelled to tropical countries is an exclusion criterion.

We excluded from the study participants who travelled to tropical countries because of the potential infections caused by consumption of unsual products.  The factors that could modify the microbiota has been extended in the row 85.

- Point 2.2. Stool sample collection should indicate also “and PCR assays “. Here, authors should explain in more detail how they have done the conversion from “the number of sequence´s copies amplified” to “bacteria count per gram”. Usually, in a real time PCR reaction, data are obtained as C(t) (cycle threshold), that is the number of cycles required for the fluorescent signal to cross the threshold. So, please clarify and explain.

We have updated this method explanation in row 123. The real time-PCR results were recalculated to bacteria count per gram. The amplifi-cation efficacy in all assays was higher than 90%. The standard curve showed a linear range across at least 5 logs of DNA concentrations with a correlation coefficient >0.99. The lowest detection limits of all assays were as low as 10–100 copies of specific bacterial 16S rDNA per reaction, which corresponds to 104—105 copies per gram of wet-weight feces.  Knowing the values of the standards and their C(t) (cycle threshold), the obtained data was converted using the right coefficients. This method and standards were developed at Herborn and used in:

 Schwiertz, A., Taras, D., Schäfer, K., Beijer, S., Bos, N. A., Donus, C., & Hardt, P. D. (2010). Microbiota and SCFA in lean and overweight healthy subjects. Obesity, 18(1), 190-195.

Schwiertz, A., Jacobi, M., Frick, J. S., Richter, M., Rusch, K., & Köhler, H. (2010). Microbiota in pediatric inflammatory bowel disease. The Journal of pediatrics, 157(2), 240-244.

Reiss, A., Jacobi, M., Rusch, K., & Schwiertz, A. (2016). Association of dietary type with fecal microbiota and short chain fatty acids in vegans and omnivores. J Int Soc Microbiota, 1, 1.

If we should explain it in more detailed way please let us know.

- In the results section: why did the authors choose the species of bacteria indicated in table 3 and not others? Are they all carbohydrate fermenting bacteria? Also Akkermansia? Why weren't protein fermenting bacteria also measured? It would be interesting to measure both groups of bacteria, carbohydrate and protein fermenting bacteria and establish a balance between beneficial and harmful bacteria. I suggest that the authors do so.

In order to select the most effective, health-promoting probiotics for bodybuilders on high-protein diet, we performed qualitative and quantitative analyzes of: protective microorganisms (anaerobic bacteria of the genus Bacteroides and Bifidobacterium, lactobacilli of the genus Lactobacillus), immunostimulatory bacteria (Enterococcus and E. coli), bacteria nourishing the intestinal epithelium (Faecalibacterium prausnitzii and Akkermansia muciniphila) and proteolytic bacteria (Clostridium, Enterobacteriaceae family including Klebsiella spp. Enterobacter spp., Citrobacter spp., Pseudomonas) and total bacteria count.

We performed genetic analyzes for non-cultivated bacteria and cultures for proteolytic bacteria. As currently only the molecular methods for the identification of bacteria are respected in the science, we have made a decision not to include all the results obtained.

- Presentation of results: authors only use tables to present their data. Could they use any other system, such as diagrams? Results are more attractive for the readers with those type of presentation.

Some data has been presented as diagram (Figure 1) in row 168.

Reviewer 2 Report

This manuscript reports the investigation of whether a high protein diet used by bodybuilders will modulate abundance of health-promoting gut bacteria. The authors assessed the differences in the abundance of select gut bacteria and fecal pH between a group of amateur bodybuilders and a control group with standard diet, and found no significant differences in the selected bacterial composition, but higher feces pH in the bodybuilder group. However, there are several key flaws in this manuscript.

  1. The authors failed to separate the effects of diet from that of exercise on gut microbiome.
  2. No clear description was reported on how energy and macronutrients intake were calculated in this manuscript. How much protein (g/kg) did participants consume every day?
  3. The author should explain why Faecalibacterium prausnitzi, Akkermansia muciniphila, Bifidobacterium infantis and Bacteroides fragilis were selected for this study. These are all carbohydrate-fermenting bacteria.
  4. Excessive protein intake will affect the abundance of protein-fermenting bacteria such as Clostridium, Bacillus, Staphylococcus, and Proteobacteria family. The authors should have included these genus/phylum of bacteria into the study.

Author Response

Answers to Reviewer 2

  1. The authors failed to separate the effects of diet from that of exercise on gut microbiome.
  1. It is difficult to separate these effects because there are too many external factors that could not be controlled (like full history of nutrition or some enviromental factors). The participants would have to live in the same place and eat exact the same products and use the same culinary techniques. Further research is necessary to explain in detail all the mechanisms that modify the microbiota and host’s health.
  1. No clear description was reported on how energy and macronutrients intake were calculated in this manuscript. How much protein (g/kg) did participants consume every day?
  1. Study subjects self-reported the estimated meals composition and weight of the used products in the diary. The mean protein consumption has been calculated and expressed as percents of dietary calorie intake and as grams per kilogram of body weight.

Bodybuilders’ protein intake (%) was significantly higher than that of control group (Mean ± SD: 33,6% ± 6,5% vs. 22% ± 6,3%; p<0,05) and exceeded the recommended 25–30% of daily energy intake from protein [11]. Still, the mean protein consumption expressed as g/kg b.w. in bodybuilders group fitted the recommended ranges and was not as high as expected and did not differ significantly from that of control group (Mean ± SD: 2,1 ± 1,5 vs. 1,7 ± 1,0%; p<0,39).

  1. The author should explain why Faecalibacterium prausnitzi, Akkermansia muciniphila, Bifidobacterium infantis and Bacteroides fragilis were selected for this study. These are all carbohydrate-fermenting bacteria.
  2. Excessive protein intake will affect the abundance of protein-fermenting bacteria such as Clostridium, Bacillus, Staphylococcus, and Proteobacteria family. The authors should have included these genus/phylum of bacteria into the study.

3 and 4. In order to select the most effective, health-promoting probiotics for bodybuilders on high-protein diet, we performed qualitative and quantitative analyzes of: protective microorganisms (anaerobic bacteria of the genus Bacteroides and Bifidobacterium, lactobacilli of the genus Lactobacillus), immunostimulatory bacteria (Enterococcus and E. coli), bacteria nourishing the intestinal epithelium (Faecalibacterium prausnitzii and Akkermansia muciniphila) and proteolytic bacteria (Clostridium, Enterobacteriaceae family including Klebsiella spp. Enterobacter spp., Citrobacter spp., Pseudomonas) and total bacteria count.

We performed genetic analyzes for non-cultivated bacteria and cultures for proteolytic bacteria. As currently only the molecular methods for the identification of bacteria are respected in the science, we have made a decision not to include all the results obtained.

Round 2

Reviewer 1 Report

The authors have done all the suggested modifications  so, from my point of view, the article is suitable for publication

Reviewer 2 Report

The authors have done the suggested modifications. It is ok for publication.